# AC⊕DC search: behind the winning solution to the FlyWire graph-matching challenge

**Daniel D. Lee**                                                                    *ddl46@cornell.edu*
*Cornell Tech, School of Electrical and Computer Engineering*
*2 W Loop Rd, New York, NY 10044*

**Arie Matsliah**                                                                    *arie@princeton.edu*
*Princeton Neuroscience Institute*
*40 Woodlands Way, Princeton, NJ 08540*

**Lawrence K. Saul**                                                          *lsaul@flatironinstitute.org*
*Flatiron Institute, Center for Computational Mathematics*
*162 Fifth Avenue, New York, NY 10010*

**Reviewed on OpenReview:** *https://openreview.net/forum?id=8MjCOMyaDf*

## Abstract

This paper describes the **A**lternating **C**ontinuous and **D**iscrete **C**ombinatorial (AC⊕DC) optimizations behind the winning solution to the FlyWire Ventral Nerve Cord Matching Challenge. The challenge was organized by the Princeton Neuroscience Institute and held over three months, ending on January 31, 2025. During this period, the challenge attracted teams of researchers with expertise in machine learning, high-performance computing, graph data mining, biological network analysis, and quadratic assignment problems. The goal of the challenge was to align the connectomes of a male and female fruit fly, and more specifically, to determine a one-to-one correspondence between the neurons in their ventral nerve cords. The connectomes were represented as sparse weighted graphs with thousands of nodes and millions of edges, and the challenge was to find the permutation that best maps the nodes and edges of one graph onto those of the other. The winning solution to the challenge alternated between two complementary approaches to graph matching—the first, a combinatorial optimization over the symmetric group of permutations, and the second, a continuous relaxation of this problem to the space of doubly stochastic matrices. For the latter, the doubly stochastic matrices were optimized by combining Frank-Wolfe methods with a fast preconditioner to solve the linear assignment problem at each iteration. We provide a complete implementation of these methods with a few hundred lines of code in MATLAB. Notably, this implementation obtains a winning score to the challenge in less than 10 minutes on a laptop computer.

## 1 Introduction

The problem of graph matching is to find the permutation that best maps the nodes and edges of one graph onto those of another. The problem arises in many areas of science and engineering where graphs are used to encode similarity, co-dependence, or the flow of information; specific applications include keypoint matching in computer vision (Conte et al., 2004; Haller et al., 2022; Zhou et al., 2023), biological and social network alignment (Chen et al., 2016; Mamano & Hayes, 2017; Lázaro et al., 2025), and vulnerability detection in software systems (Li et al., 2019). The problem is solved in practice by maximizing an objective function that scores each permutation by quantifying the similarity of nodes and edges that it brings into correspondence. Graph matching is in general an NP-hard problem; one can find a globally optimal solution by exhaustively

considering all permutations, but this is only possible for small graphs. For large graphs, the best solvers rely on approximate search algorithms that attempt to find a high-scoring match.

A particular instance of this problem was at the heart of a recent challenge[1] posed by the FlyWire consortium at the Princeton Neuroscience Institute. The goal of the challenge was to align the connectomes of a male and female fruit fly, where each connectome was represented by a sparse graph. The nodes in these graphs represented neurons, and the edges indicated which neurons were connected by synapses. These graphs were also directed and weighted: each edge counted the number of synapses observed in a particular direction between two neurons. The determination of these connectomes—down to the level of individual synapses— was the culmination of many years of painstaking research (Takemura et al., 2017; Dorkenwald et al., 2024; Bates et al., 2025), and the FlyWire consortium has since issued a stream of open challenges to further process these results. This particular challenge was posed in the winter of 2024, and its purpose was to study how sex differences in fruit flies are manifested in the connectomes of their ventral nerve cords (VNCs).

There were two features of this challenge that made for an especially compelling problem in graph matching. First was the problem size: each VNC connectome was represented by a graph with 18524 nodes and millions of edges. Notably, these connectomes are larger by two orders of magnitude than that of the hermaphrodite *C. elegans*, which consists of only 302 neurons (Chen et al., 2016; Varshney et al., 2011; Lázaro et al., 2025). The graphs in the challenge were also much larger than those of typical benchmarks for keypoint-matching problems in computer vision (Haller et al., 2022) and other related problems in graph-representation learning (Morris et al., 2020). At the same time, the VNC connectomes were small enough that soft matches, represented by dense doubly stochastic matrices, could fit into the memory of a modestly equipped computer. It was therefore possible to explore approaches that took advantage of this ability and did not rely on purely combinatorial techniques.

The second distinguishing feature of the challenge was the particular way that different matches were scored. Let $\pi$ denote the permutation of the indices $\{1, 2, \ldots, n\}$, with $n = 18524$, that maps $i$ to $\pi_i$, and let $A$ and $B$ denote, respectively, the sparse $n \times n$ matrices whose nonzero elements record edge weights in the male and female connectomes. The alignment score for the VNC matching challenge was computed as

$$\mathcal{S}(\pi) = \sum_{ij} \min\big(A_{ij}, B_{\pi_i \pi_j}\big). \tag{1}$$

The weighted-intersection score in eq. (1) is a variant on the weighted Jaccard distance, and it was chosen by the challenge organizers because it appeared to correlate better with known biological isomorphisms. In particular, they found that eq. (1) was more robust with respect to these isomorphisms than scores based on simple correlation or cosine distance. The score in eq. (1) can be computed efficiently by restricting the sum to nonzero elements of $A$ and $B$.

The challenge ran for three months, and it attracted teams of researchers with expertise in machine learning, high-performance computing, graph data mining, biological network analysis, and quadratic assignment problems. The challenge organizers provided a baseline match that was determined from neuron cell types, and this benchmark solution (though not the cell type metadata) was made available for teams to use as a warm start. Nearly all teams continued to submit improved solutions up to the deadline on January 31, 2025, and the scores of these submissions were independently verified by the challenge organizers. Throughout the challenge, scores were not disclosed, and different teams did not share code or details of their solutions.

In this paper, we reveal the methods behind the winning solution to the challenge. The solution combined two complementary approaches to graph matching: the first was a combinatorial optimization over the space of permutation matrices, and the second was a continuous relaxation of this problem to the space of doubly stochastic matrices. Both of these approaches were pursued individually by other high-placing teams in the challenge—teams which included, for example, the co-inventors of leading methods based on simulated annealing (Mamano & Hayes, 2017) and fast approximate quadratic programming (Vogelstein et al., 2015). These methods, based on *either* discrete or continuous search, provided strong baselines for the challenge, placing among the top five scores on the FlyWire VNC leaderboard. However, as we describe here, it was

---

[1]https://codex.flywire.ai/app/vnc_matching_challenge

a combination of these complementary approaches that led much more quickly to a winning solution. We refer to this alternation of continuous and discrete combinatorial methods as AC⊕DC search.

The continuous relaxation of this problem is obtained by extending the score in eq. (1) to the convex set of doubly stochastic matrices. The relaxed objective is given by

$$\mathcal{S}(P) = \sum_{ijk\ell} \min(A_{ij}, B_{k\ell}) P_{ik} P_{j\ell}, \tag{2}$$

where $P$ is an $n \times n$ nonnegative matrix whose rows and columns sum to one. Note that the objective in eq. (2) is quadratic *but not concave* in its argument; also, when $P$ is dense, it appears naively to require $O(n^4)$ operations to perform its quadruple sum. The winning solution optimized eq. (2) by adapting Frank-Wolfe methods for constrained convex optimization (Frank & Wolfe, 1956) alongside a fast preconditioner to solve the linear assignment problem at each iteration. This type of relaxation has been used successfully for other quadratic assignment problems (Vogelstein et al., 2015). The main novelties described in this paper are highly optimized routines for computing the gradient of eq. (2) and projecting this gradient into the space of permutation matrices.

With these techniques, we show how to obtain a winning score to the challenge in under 10 minutes, on a laptop computer, with a few hundred lines of code in MATLAB. Though not revealed at the time—because teams on the leaderboard were listed by the date of their most recent submission—the AC⊕DC methods held the top score for the final forty days of the challenge. These methods should be of direct interest to other researchers in biological network analysis, and also of general interest to researchers in machine learning whose problems require optimizations over the permutation group (Mena et al., 2018; Cuturi et al., 2019; Grover et al., 2019; Santa Cruz et al., 2019; Prillo & Eisenschlos, 2020; Dröge et al., 2023).

The organization of this paper is as follows. In section 2, we describe a greedy search algorithm for the combinatorial optimization of eq. (1) over the space of permutation matrices. This approach has the advantage of simplicity, and it also directly optimizes the score in eq. (1). But it improves the score slowly in the early stages of optimization, and in the later stages, it is prone to getting stuck. In section 3, we describe a first-order method for the continuous optimization of eq. (2). This method has the advantage that it makes rapid initial progress, but it does not converge to a permutation matrix that maximizes eq. (1). In section 4, we describe the winning solution that is found by alternating these approaches, and we also discuss further techniques (albeit with diminishing returns) for optimizing the scores in eqs. (1) and (2). Finally, in section 5, we conclude with a discussion of open problems and directions for future work.

## 2 Discrete search

There are many ways to search for a permutation that maximizes the score in eq. (1). Arguably the simplest is a hill-climbing approach that makes local moves in the space of $n!$ permutations. This approach can equivalently be viewed as an optimization over the space of $n \times n$ permutation matrices—that is, matrices whose elements are equal to zero or one and whose rows and columns sum to one. Within this approach, there are also many types of local moves that can be considered, but the simplest are those that swap exactly one pair of indices (viewing permutations as shuffles) or exactly one pair of rows (viewing them as matrices). In this section, we describe an efficient way to evaluate these moves and illustrate the strengths and weaknesses of this approach.

### 2.1 Evaluation of pairwise swaps

Our first goal is to evaluate how the score in eq. (1) is changed by single pairwise swaps. The following notation will be useful. Let $P^\pi$ denote the $n \times n$ permutation matrix corresponding to the permutation $\pi$, whose elements are given by $P_{ik}^\pi = \delta(\pi_i, k)$, where $\delta(\cdot, \cdot)$ is the Kronecker delta function. Similarly, let $\sigma_{ij}$ denote the permutation that swaps $i$ and $j$ while leaving all other indices intact, and let $\pi \circ \sigma_{ij}$ denote the composition (from right to left) of these two permutations—that is, the permutation obtained by first swapping $i$ and $j$ and then permuting the indices according to $\pi$.

We begin by computing the $n \times n$ symmetric matrix $\Delta^\pi$ whose elements record the difference in scores between permutations that are related by a single pairwise swap of indices. In particular, let

$$\Delta_{ij}^\pi = \mathcal{S}(\pi \circ \sigma_{ij}) - \mathcal{S}(\pi), \tag{3}$$

so that the diagonal elements of $\Delta^\pi$ are zero, while the nonzero elements indicate those local moves in the space of permutations that change the score. Note that for this challenge, with $n = 18524$ nodes per graph, each matrix $\Delta^\pi$ records the effect of over 171 million pairwise swaps.

Let $\mathcal{P}$ denote the convex set of doubly stochastic $n \times n$ matrices. Since the score in eq. (2) is quadratic in the matrix $P \in \mathcal{P}$, the differences in eq. (3) can also be expressed in terms of the gradient and Hessian of this score. The gradient of eq. (2) is given by the $n \times n$ matrix with elements

$$\big[ \nabla \mathcal{S}(P) \big]_{j\ell} = \sum_{ik} \Big[ \min(A_{ij}, B_{k\ell}) + \min(A_{ji}, B_{\ell k}) \Big] P_{ik}, \tag{4}$$

and it is generally a *dense* matrix even when the matrices $A$, $B$, and $P$ in eq. (4) are sparse. Of particular interest is the form of this gradient at the permutation matrix $P^\pi$. We denote this gradient by $G^\pi = \nabla \mathcal{S}(P^\pi)$, and its elements are given by

$$G_{j\ell}^\pi = \sum_i \Big[ \min(A_{ij}, B_{\pi_i \ell}) + \min(A_{ji}, B_{\ell \pi_i}) \Big]. \tag{5}$$

The gradient in eq. (5) can be computed in $O(n^2)$ operations by exploiting the sparsity of the connectome weights in the matrices $A$ and $B$. In particular, for the matrices of size $n = 18524$ in this challenge, this gradient takes about 12 seconds to compute on a Macbook Pro (M1 Max) laptop.

Next we consider the way in which the Hessian of the score in eq. (2) enters into the calculation of differences in eq. (3). To this end, we introduce a new matrix $B^\pi$, whose elements are obtained by permuting the rows and columns of $B$ according to the permutation $\pi$; in particular,

$$B_{ij}^\pi = B_{\pi_i \pi_j}. \tag{6}$$

The elements of $B^\pi$ appear in certain combinations of Hessian elements that arise repeatedly in the calculation of the matrix $\Delta^\pi$. As further shorthand, we define the functions

$$H_{ij}^\pi(a) = \min(a, B_{ii}^\pi) + \min(a, B_{jj}^\pi) - \min(a, B_{ij}^\pi) - \min(a, B_{ji}^\pi), \tag{7}$$

where in practice we will take the argument $a$ to be a particular connectome weight from the matrix $A$. For example, when $a = A_{ii}$, the right side of eq. (7) expresses a particular linear combination of elements from the Hessian of eq. (2) evaluated at the matrix $P^\pi$.

With the above definitions, we can express the effects of swaps in eq. (3) in terms of the connectome weights and the gradient and Hessian of the score. In terms of these quantities, the elements of $\Delta^\pi$ are given by

$$\Delta_{ij}^\pi = G_{i\pi_j}^\pi + G_{j\pi_i}^\pi - G_{i\pi_i}^\pi - G_{j\pi_j}^\pi + H_{ij}^\pi(A_{ii}) + H_{ij}^\pi(A_{jj}) - H_{ij}^\pi(A_{ij}) - H_{ij}^\pi(A_{ji}), \tag{8}$$

and again, all $n^2$ matrix elements in this equation can be computed in $O(n^2)$ operations for a given permutation $\pi$. For the matrices of size $n = 18524$ in this challenge, it takes about 3 additional seconds to compute the elements in eq. (8) on top of the gradient in eq. (5). We provide pseudocode for a procedure (EVALUATESWAPS) to compute these elements in Algorithm 1.

It is possible for none of the matrix elements $\Delta_{ij}^\pi$ in eq. (8) to be positive. When this is the case, it indicates that the permutation $\pi$ cannot be improved by a single pairwise swap of indices. Otherwise, the largest (i.e., most positive) element of $\Delta^\pi$ indicates the swap that most improves the weighted-intersection score in eq. (1). This suggests a simple algorithm for greedy local search which we describe in the next section.

---

**Algorithm 1** Given connectome weights $A, B \in \mathbb{R}^{n \times n}$ and a base permutation $\pi$, evaluate the score differences in eq. (8) obtained by a single pairwise swap of indices.

---

**procedure** $\Delta = \text{EVALUATESWAPS}(A,B,\pi)$
    ▷ *Compute gradient at $\pi$ and permute rows and columns of B*
    **for** $i \leftarrow 1$ to $n$ **do**
        **for** $j \leftarrow 1$ to $n$ **do**
            $G_{ij} \leftarrow \sum_{k=1}^{n}[\min(A_{ki}, B_{\pi_k j}) + \min(A_{ik}, B_{j\pi_k})]$
            $B_{ij}^{\pi} \leftarrow B_{\pi_i \pi_j}$
        **end for**
    **end for**
    ▷ *Evaluate difference in scores due to pairwise swaps*
    **for** $i \leftarrow 1$ to $n$ **do**
        **for** $j \leftarrow 1$ to $n$ **do**
            $H_{ii} \leftarrow \min(A_{ii}, B_{ii}^{\pi}) + \min(A_{ii}, B_{jj}^{\pi}) - \min(A_{ii}, B_{ij}^{\pi}) - \min(A_{ii}, B_{ji}^{\pi})$
            $H_{jj} \leftarrow \min(A_{jj}, B_{ii}^{\pi}) + \min(A_{jj}, B_{jj}^{\pi}) - \min(A_{jj}, B_{ij}^{\pi}) - \min(A_{jj}, B_{ji}^{\pi})$
            $H_{ij} \leftarrow \min(A_{ij}, B_{ii}^{\pi}) + \min(A_{ij}, B_{jj}^{\pi}) - \min(A_{ij}, B_{ij}^{\pi}) - \min(A_{ij}, B_{ji}^{\pi})$
            $H_{ji} \leftarrow \min(A_{ji}, B_{ii}^{\pi}) + \min(A_{ji}, B_{jj}^{\pi}) - \min(A_{ji}, B_{ij}^{\pi}) - \min(A_{ji}, B_{ji}^{\pi})$
            $\Delta_{ij} \leftarrow G_{i\pi_j} + G_{j\pi_i} - G_{i\pi_i} - G_{j\pi_j} + H_{ii} + H_{jj} - H_{ij} - H_{ji}$
        **end for**
    **end for**
**end procedure**

---

## 2.2 Greedy search with pairwise swaps

Starting from an initial permutation $\pi$, one can attempt to optimize the score in eq. (1) by alternating two procedures; the first evaluates the effect of each pairwise swap by computing its corresponding element in $\Delta^{\pi}$, and the second performs those swaps that seem likely to increase the score by the largest amount. We give the pseudocode for a greedy search based on these two procedures in Algorithm 2. The search terminates when the first procedure, sketched in Algorithm 1, returns a matrix $\Delta^{\pi}$ with no positive elements.

We use a threshold $\tau$ to adjust the balance of time spent in these two procedures. The MAKESWAPS procedure in Algorithm 2 performs up to $\tau$ swaps that increase the score while skipping over swaps that do not. The threshold is only needed in the early stages of optimization, when the permutation $\pi$ is very far from optimal; in this case, the matrix $\Delta^{\pi}$ returned by the first procedure (EVALUATESWAPS) may have an inordinately large number of positive elements. The MAKESWAPS procedure considers pairwise swaps in descending order of their corresponding elements in $\Delta^{\pi}$. If the maximal element of $\Delta^{\pi}$ is positive, then the first such swap is guaranteed to yield a permutation with a higher score. However, successive swaps are *not* guaranteed to increase the score, even if they correspond to positive elements of $\Delta^{\pi}$, due to possible interference with previously executed swaps. (A trivial example of such interference arises from the symmetry of the matrix $\Delta^{\pi}$: a swap $j \leftrightarrow i$ will exactly negate the gain from an immediately preceding swap $i \leftrightarrow j$.)

The left panel of Fig. 1 shows results from the greedy search in Algorithm 2 for different thresholds on the maximum numbers of pairwise swaps per iteration. All of these runs were initialized from the benchmark solution with score 5154247 provided by the challenge organizers. The algorithm converges to different solutions for different thresholds $\tau$, but all of these solutions have scores around 5818K. These solutions are evidence of the large number of local maxima in this problem: there are many permutations whose scores cannot be improved by any pairwise swaps of indices (of which there are over 171 million).

There are many ways to augment the greedy search in Algorithm 2 so that it discovers higher-scoring permutations. One is to introduce an element of randomness, sometimes performing a pairwise swap that decreases the score, as is done in simulated annealing (Mamano & Hayes, 2017). Another is to evaluate and perform higher-order moves that swap three or more indices at a time. While these approaches may require more resources, one can also optimize them aggressively, in faster languages than MATLAB, while exploiting opportunities for parallelism (e.g., multi-core, GPUs) (Koblentz, 2025).

---

**Algorithm 2** Given connectome weights $A, B \in \mathbb{R}^{n \times n}$ and an initial permutation $\pi_0$, perform a greedy search with up to $\tau$ pairwise swaps (per iteration) to find a local maximum in the alignment score of eq. (1).

---

**procedure** $\pi = \text{GREEDYSEARCH}(A,B,\pi_0,\tau)$
    $\pi \leftarrow \pi_0$
    $\Delta \leftarrow \text{EVALUATESWAPS}(A, B, \pi)$
    **while** $(\max_{ij}(\Delta_{ij}) > 0)$ **do**
        $\pi \leftarrow \text{MAKESWAPS}(A, B, \pi, \Delta, \tau)$
        $\Delta \leftarrow \text{EVALUATESWAPS}(A, B, \pi)$
    **end while**
**end procedure**

**procedure** $\pi = \text{MAKESWAPS}(A,B,\pi,\Delta,\tau)$
    $\mathcal{S} \leftarrow \sum_{ij} \min(A_{ij}, B_{\pi_i \pi_j})$
    **while** $((\tau > 0) \text{ AND } (\max_{ij}(\Delta_{ij}) > 0))$ **do**
        $(i, j) \leftarrow \text{argmax}_{ij}(\Delta_{ij})$
        $\pi' \leftarrow \pi \circ \sigma_{ij}$
        $\mathcal{S}' \leftarrow \sum_{ij} \min(A_{ij}, B_{\pi'_i \pi'_j})$
        **if** $(\mathcal{S}' > \mathcal{S})$ **then**
            $(\pi, \mathcal{S}, \tau) \leftarrow (\pi', \mathcal{S}', \tau - 1)$
        **end if**
        $(\Delta_{ij}, \Delta_{ji}) \leftarrow (0, 0)$
    **end while**
**end procedure**

---

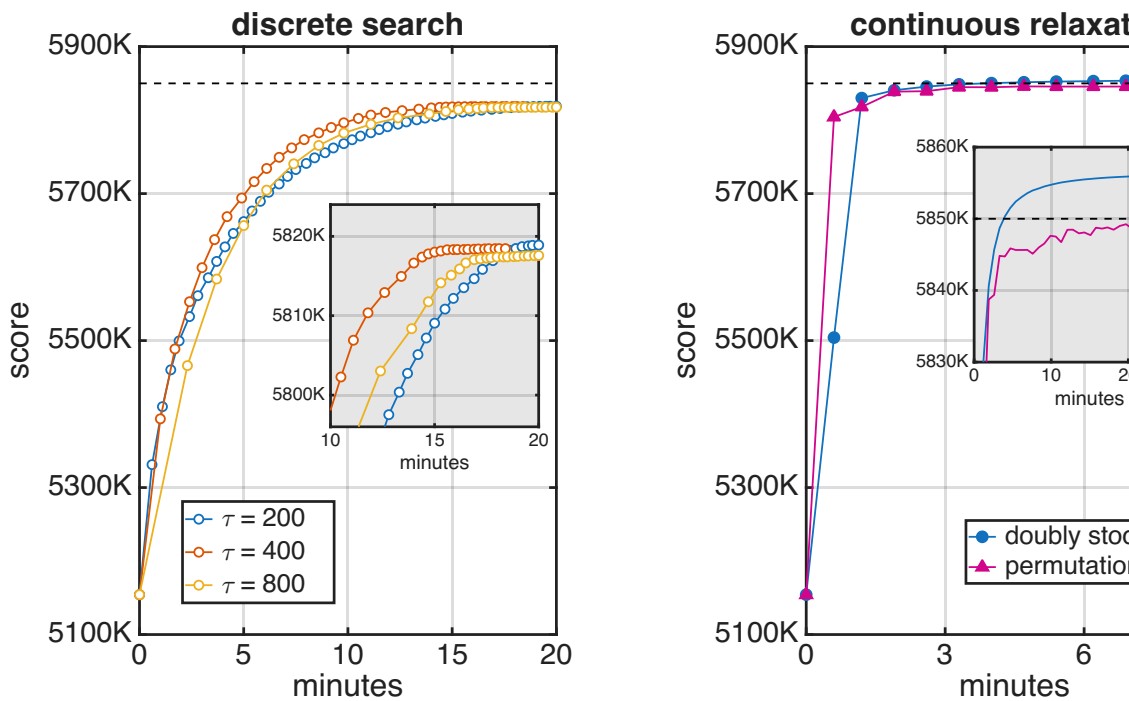

Figure 1: Alignment scores in eqs. (1–2) versus wall clock time starting from the benchmark solution at score 5154247. *Left.* Greedy discrete search in Algorithm 2 with up to $\tau$ pairwise swaps per iteration. *Right.* Frank-Wolfe updates in eqs. (11) and (13) to optimize the continuous relaxation in eq. (2). Neither method by itself converges to a winning score for the challenge (indicated by the dashed line at 5850K).

Multiple teams experimented with these ideas in the days and weeks leading up to the deadline. But even with considerably longer runs, these more elaborate forms of discrete search were not able to reach the dashed line in Fig. 1, indicating a score (at 5850K) that was high enough to win the challenge. As mentioned earlier, however, this winning score can be obtained in under 10 minutes by combining discrete and continuous approaches to the optimization of eq. (1). With this goal in mind, we now turn to the latter approach.

# 3 Continuous relaxation

In this section we describe a complementary approach to graph matching, one based on a continuous optimization over the convex set of doubly stochastic matrices. This type of relaxation has been studied previously for quadratic assignment problems (Vogelstein et al., 2015), but for the best results this continuous optimization must be tailored specifically to the weighted-intersection score in eq. (2). This score is a quadratic function of $P$, but it is not concave, and therefore an iterative hill-climbing procedure is not guaranteed to find its global maximum. Here we show that an iterative procedure, based on the Frank-Wolfe algorithm for constrained convex optimization (Frank & Wolfe, 1956), can be adapted to this problem with extremely competitive results. One part of this procedure is to calculate a projected gradient, and we also describe a fast preconditioner for this calculation that is crucial to its overall efficiency.

## 3.1 Frank-Wolfe updates

The iterative procedure is shown in Algorithm 3, and it alternates between three steps. The first step computes the gradient in eq. (4). As mentioned previously, it takes about 12 sec to compute this gradient at a permutation matrix of size $n = 18524$. For this iterative procedure, we need to compute the gradient at doubly stochastic matrices, which in general can take much longer. As we shall see, however, the procedure converges very quickly, so that in practice—*if the search is initialized by a permutation matrix*—we only need to compute gradients for doubly stochastic matrices that are highly sparse. When this is the case, it takes only slightly longer to compute the gradient in eq. (4).

The second step of the iterative procedure projects this gradient back into the convex set of doubly stochastic matrices. In particular, this step computes

$$Q_t = \underset{Q \in \mathcal{P}}{\operatorname{argmax}} \left( \operatorname{trace}\left[ \nabla \mathcal{S}(P_t)^\top Q \right] \right). \tag{9}$$

Note that eq. (9) defines a linear program whose solution always lies at a *vertex* of the set $\mathcal{P}$; in other words, its solution $Q_t$ is not merely a doubly stochastic matrix, but also a permutation matrix. The optimization in eq. (9) is most commonly known as the linear assignment problem, or the problem of perfect matching in a complete bipartite graph. It can be solved by the so-called Hungarian method (Kuhn, 1955) in polynomial time (Munkres, 1957; Edmonds & Karp, 1972; Tomizawa, 1971), but this is not the approach that was used in the winning solution to the challenge. We discuss this step more later, since without further acceleration it presents the biggest computational bottleneck.

The third step of the iterative procedure is to find the convex combination of $P_t$ and $Q_t$ that maximizes the score in eq. (2). In particular, the update is given by

$$\alpha_t = \underset{\alpha \in [0,1]}{\operatorname{argmax}} \left[ \mathcal{S}\big((1-\alpha)P_t + \alpha Q_t\big) \right], \tag{10}$$

$$P_{t+1} = (1-\alpha_t)P_t + \alpha_t Q_t. \tag{11}$$

In practice, it is not necessary to perform a line search to compute the optimal convex combination in eq. (10). Instead one can simply calculate the point where the gradient of the score vanishes along the line connecting $P_t$ and $Q_t$. Since the score in eq. (2) is quadratic in its argument, this gradient vanishes at some point $(1-\lambda)P_t + \lambda Q_t$ where $\lambda \in \mathbb{R}$. In particular, $\lambda$ satisfies the linear equation

$$(1-\lambda) \operatorname{trace}\left[ (Q_t - P_t)^\top \nabla \mathcal{S}(P_t) \right] = -\lambda \operatorname{trace}\left[ (Q_t - P_t)^\top \nabla \mathcal{S}(Q_t) \right]. \tag{12}$$

If $\lambda \in [0,1]$, then the weight $\alpha_t$ in eq. (10) is simply equal to $\lambda$. If $\lambda \notin [0,1]$, then there are two possibilities: either the score along the line from $P_t$ to $Q_t$ is *concave* with a *maximum* at $\lambda > 1$, or it is *convex* with a *minimum* at $\lambda < 0$. In both these cases, eq. (10) yields $\alpha_t = 1$. Finally, we note that the updates in eqs. (9–11) converge monotonically to a doubly stochastic matrix that is a stationary point (where the gradient has no component inside $\mathcal{P}$) of this procedure.

---

**Algorithm 3** Given connectome weights $A, B \in \mathbb{R}^{n \times n}$ and an initial doubly stochastic matrix $P_0$, perform $T$ Frank-Wolfe updates to optimize the score in eq. (2), then return the doubly stochastic matrix $P_T$ and permutation matrix $\Pi_T$ found from these updates.

---

> **procedure** $(P_T, \Pi_T) = \text{DoFrankWolfe}(A, B, P_0, T)$
>     $P \leftarrow P_0$
>     **for** $t \leftarrow 1$ to $T$ **do**
>         ▷ *Compute gradient*
>         **for** $i \leftarrow 1$ to $n$ **do**
>             **for** $j \leftarrow 1$ to $n$ **do**
>                 $G_{ij} \leftarrow \sum_{k\ell} \big[ \min(A_{ki}, B_{\ell j}) + \min(A_{ik}, B_{j\ell}) \big] P_{k\ell}$
>             **end for**
>         **end for**
>         ▷ *Project gradient, compute step size, and interpolate*
>         $Q \leftarrow \text{argmax}_{Q \in \mathcal{P}} \, \text{trace}\big[G^\top Q\big]$
>         $\alpha \leftarrow \text{argmax}_{\alpha \in [0,1]} \big[ \mathcal{S}((1-\alpha)P + \alpha Q) \big]$
>         $P \leftarrow (1-\alpha)P + \alpha Q$
>         ▷ *Compute closest-matching permutation matrix*
>         $\Pi \leftarrow \text{argmax}_{\Pi \in \mathcal{P}} \, \text{trace}\big[P^\top \Pi\big]$
>     **end for**
>     $P_T \leftarrow P$
>     $\Pi_T \leftarrow \Pi$
> **end procedure**

---

## 3.2 Application to graph matching

While the Frank-Wolfe updates lead to monotonic improvement in the score of eq. (2), they converge in general to a doubly stochastic matrix and not a permutation matrix. But it is the latter that is needed to align two graphs with a score given by eq. (1). To rectify this problem, we also compute a permutation matrix $\Pi_t$ at each iteration of the updates in eqs. (9–11). This is done by projecting the doubly stochastic matrix $P_t$ into the space of permutation matrices:

$$\Pi_t = \underset{\Pi \in \mathcal{P}}{\text{argmax}} \left( \text{trace}\left[ P_t^\top \Pi \right] \right). \tag{13}$$

Eq. (13) is a linear program whose solution is the closest-matching permutation matrix to $P_t$. Again this can be solved by the Hungarian method or any other algorithm for perfect matching in a complete bipartite graph. In practice the linear program in eq. (13) is much faster to solve than the one in eq. (9); the reason is that the doubly stochastic matrix $P_t$ in eq. (13) is highly sparse—expressible as a convex combination of a small number of permutation matrices—whereas the gradient $\nabla \mathcal{S}(P_t)$ in eq. (9) is dense.

Since the updates for $P_t$ in eqs. (9–11) converge to a point inside the convex set of doubly stochastic matrices, it is also true that their projections to $\Pi_t$ in eq. (13) converge to a permutation matrix at a vertex of this set. But while the scores $\{\mathcal{S}(P_t)\}_{t=0}^T$ of these doubly stochastic matrices increase monotonically as a result of these updates, the same is *not* true for the scores $\{\mathcal{S}(\Pi_t)\}_{t=0}^T$ of their closest-matching permutation matrices. The right panel of Fig. 1 plots the scores from these updates starting from the benchmark solution with score 5154247.

From the results in Fig. 1, we make several observations of interest. First, at the outset of the optimization, the continuous updates in the convex set of doubly stochastic matrices (shown right) increase the score much more rapidly than the discrete search based on pairwise swaps (shown left). Second, the scores of the permutation matrices $\Pi_t$ in eq. (13) generally track the scores of the doubly stochastic matrices $P_t$ in eq. (11), but the latter increase monotonically while the former do not. Third, the scores of the doubly stochastic matrices $P_t$ saturate around 5856K, while those of the permutation matrices $\Pi_t$ saturate just below 5850K. In particular, these updates by themselves do not obtain a handily winning score for the challenge.

### 3.3 Acceleration by preconditioning

In this section we describe how the winning entry to the challenge solved the linear program in eq. (9). As mentioned previously, this problem is equivalent to one of perfect matching, and it is more typically posed in terms of a cost matrix, $C \in \mathbb{R}^{n \times n}$, where the goal is to find the permutation $\pi$ that minimizes the linear assignment cost

$$\text{trace}(C^\top P^\pi) = \sum_i C_{i\pi_i}. \tag{14}$$

There is an internal (though not especially well-documented) routine in MATLAB that solves this problem by permuting large entries to the diagonal of a sparse matrix (Duff & Koster, 2001). It assumes that $C$ is stored as a dense matrix, and it is called as

$$\pi = \texttt{matlab.internal.graph.perfectMatching}(C). \tag{15}$$

We used this internal routine to solve the linear programs in eqs. (9) and (13) whose cost matrices had $n = 18524$ rows and columns. The routine is based on a polynomial-time algorithm, but it can be very slow if called in the above manner when $C$ is a dense matrix. For example, when $C = -\nabla\mathcal{S}(P_t)$, this routine requires 10 to 15 minutes per call on a MacBook Pro (M1 Max) with 64 GB of RAM. Of course it would not be possible to obtain a winning solution in less than 15 minutes if each iteration of Algorithm 3 required this much computation.

We discovered a heuristic that greatly accelerates this routine for perfect matching when it is called with the gradients $\nabla\mathcal{S}(P_t)$ that appear in eq. (9). The heuristic is based on three observations. First, the result in eq. (15) is unaffected if we shift any row or column of the cost matrix by a constant value. Second, the result is trivially equal to the identity permutation if $C$ has negative elements on the diagonal and nonnegative elements off the diagonal. Third, suppose that an approximate solution $\omega$ can be guessed for eq. (15), where $\omega$ is a permutation that nearly solves the linear assignment problem. Then the matrix product $C \cdot (P^\omega)^\top$ should be closer than $C$ to a matrix whose smallest entries appear on the diagonal.

Based on these observations, we discovered something akin to a preconditioner for the routine in eq. (15) when $C = -\nabla\mathcal{S}(P_t)$. We describe this preconditioner in detail because it yielded a significant speedup, reducing the time per call by a factor of $50\times$, or from minutes to seconds. As shorthand, let $\mathbb{1} \in \mathbb{R}^n$ denote the column vector of all ones, and let $\text{diag}(\cdot)$ denote the column vector of diagonal elements from its matrix argument. We start by observing that $\Pi_t$ in eq. (13) provides an approximate guess for $Q_t$ in eq. (9). With this and the previous observations in mind, we solve eq. (9) in the following way:

$$(\text{PERMUTE}) \quad \Lambda = \nabla\mathcal{S}(P_t)\Pi_t^\top, \tag{16}$$

$$(\text{SHIFT}) \quad \Omega = \Lambda + \text{diag}(\Lambda)\mathbb{1}^\top + \mathbb{1}\,\text{diag}(\Lambda)^\top - \mathbb{1}\mathbb{1}^\top\Lambda - \Lambda\mathbb{1}\mathbb{1}^\top, \tag{17}$$

$$(\text{MATCH}) \quad \omega = \texttt{matlab.internal.graph.perfectMatching}(-\Omega), \tag{18}$$

$$(\text{UNPERMUTE}) \quad Q_t = P^\omega\Pi_t. \tag{19}$$

Intuitively, the first of these steps (PERMUTE) ensures that $\Lambda$ has positive elements on the diagonal, the second (SHIFT) makes it more likely that $\Omega$ has negative elements off the diagonal, and the third (MATCH) is fastest when $\Omega$ has positive elements on the diagonal and none elsewhere, in which case $\omega$ is close to the identity permutation. We do not have a formal justification for this heuristic, but in practice it removed eq. (9) as the main bottleneck in Algorithm 3. Further details can be found in Appendix A.

## 4 A winning solution (and beyond)

It is possible to combine the methods for search in the last two sections and reap the advantages of both. The discrete swaps in section 2 lead to slow but steady improvement until they reach a local maximum from which they cannot escape. The continuous Frank-Wolfe updates in section 3 lead to rapid improvement in the weighted-intersection score, but they plateau when the high-scoring doubly stochastic matrices in eq. (2) do not project to high-scoring permutation matrices in eq. (1). A winning solution can be quickly obtained by alternating these approaches, using each to offset the weaknesses of the other. This is the method of alternating continuous and discrete combinatorial (AC$\oplus$DC) search.

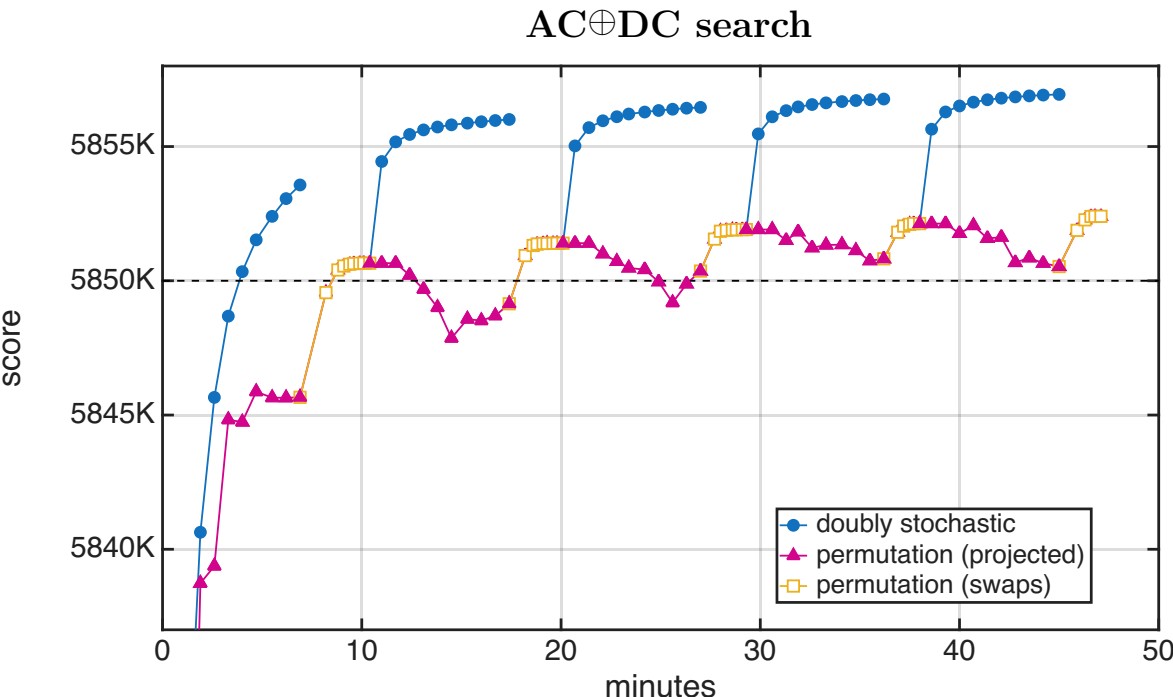

Figure 2: Alignment scores in eqs. (1–2) versus wall clock time starting from the benchmark solution at score 5154247. The scores were obtained by alternating updates for continuous and discrete combinatorial (AC⊕DC) search—in particular, Frank-Wolfe updates (in batches of ten) for the former and greedy pairwise swaps (repeated until no further swaps improved the score) for the latter. It takes less than 10 minutes for this method to produce a winning score for the challenge (indicated by the dashed line at 5850K).

### 4.1  AC⊕DC search

Fig. 2 shows the results from this alternating approach. First, we use ten Frank-Wolfe updates to climb from the benchmark score at 5154K to a score above 5845K in less than 7 minutes. Then we apply pairwise swaps until the score can no longer be further improved; in 3 additional minutes, these swaps produce a solution whose score exceeds 5850K, higher than all but the winning entry to the challenge. As shown in the figure, the score can be further improved by alternating these different types of search, with the Frank-Wolfe updates jumping out of the local maximum reached by the pairwise swaps, and the pairwise swaps reaching higher scores from wherever they are subsequently initialized. This combined approach reaches a score over 5852K in under 50 minutes. In the supplementary material, we provide MATLAB code (less than 400 lines) to obtain this winning solution, as well as code to generate all the figures in this paper.

Fig. 2 also highlights the different role played by the continuous Frank-Wolfe updates in the later stages of optimization. In the first few iterations, before the five-minute mark, there is a high degree of correlation between the scores of the doubly stochastic matrices in eq. (11) and their closest-matching permutation matrices in eq. (13): when the former increase (shown in blue), so do the latter (shown in red). But this relationship no longer holds past the five-minute mark in Fig. 2. In this regime, we see that higher-scoring interior solutions often project to lower-scoring permutation matrices. Nevertheless these continuous updates still play a crucial role: they re-initialize the next stage of discrete updates in a basin of attraction where pairwise swaps can reach a higher maximum of the score in eq. (1). The overall result is the seesaw pattern of improvement between the red and yellow curves that we see in Fig. 2. Finally we note that the experiments in Fig. 1 provide an ablation study, revealing the ceilings in performance when discrete and continuous updates are not alternated but instead used in isolation.

## 4.2 Further improvements

For the VNC matching challenge, we have shown that a score of over 5852K can be reached in under one hour by combining simple methods for discrete and continuous search. In this section, we give a brief overview of additional methods to further improve the score. At the outset, we note that above 5852K the optimization appears to enter a regime of diminishing returns. As shown in Fig. 2, it takes only a few minutes to improve the benchmark score by nearly 700K, and less than an hour after that to improve the score by an additional 10K. But beyond this regime it takes many additional hours—even for the more elaborate methods we discuss next—to obtain improvements that are orders-of-magnitude less. In light of this, we only provide a high-level sketch of these methods.

### 4.2.1 Higher-order swaps

The discrete search in Algorithm 2 quickly finds a solution that cannot be improved by further pairwise swaps. This search over permutation matrices can be extended by considering higher-order swaps that permute more than two indices at a time. For higher-order swaps, however, it is no longer feasible to evaluate all possible local moves before considering which ones to perform; there are, for instance, over one trillion different three-node swaps that can be performed in a graph with $n = 18524$ nodes. Instead one can evaluate a subset of higher-order moves that seem most likely to yield improvements. For example, we considered the subset of three-cycles $\{(i \rightarrow j \rightarrow k \rightarrow i)\}$ where the index $k$ was chosen greedily for all pairwise swaps $\{(i \leftrightarrow j)\}$ that did not reduce the score by a certain threshold. We also devised similar strategies for considering many different types of higher-order swaps. In total, our most sophisticated discrete search considered not only pairwise swaps, but also 3-cycles, 4-cycles, and 5-cycles, as well as 2x2, 3x2, 3x3, 4x2, 5x2, 2x2x2, 3x2x2, and 2x2x2x2 swaps. With these higher-order swaps, it takes another dozen hours to boost the score from 5852K to 5853K (amounting to a gain of less than 0.01%).

### 4.2.2 Multiplicative updates

The Frank-Wolfe updates in Algorithm 3 produce a sequence of doubly stochastic matrices that improve the score in eq. (2). When these updates are initialized from a permutation matrix, they produce a sequence of *sparse* doubly stochastic matrices. This sparsity has certain computational advantages: for example, it can be exploited to compute the gradient in eq. (4) much more efficiently. But it also has potential disadvantages; in particular, an optimization restricted to sparse solutions may not fully leverage the continuous search that is afforded by the relaxation to doubly stochastic matrices.

Recall that the updates in eq. (11) are *additive* updates in which the existing solution $P_t$ is linearly interpolated with the projected gradient $Q_t$. We also experimented with *multiplicative updates* that use the gradient in eq. (4) quite differently. These updates take the form

$$[P_{t+1}]_{ij} = [P_t]_{ij} \cdot \frac{[\nabla \mathcal{S}(P_t)]_{ij}}{u_i + v_j}, \tag{20}$$

where in the numerator of eq. (20) appear the elements of the gradient $\nabla \mathcal{S}(P_t)$ and in the denominator appear Lagrange multipliers $u, v \in \mathbb{R}^n$. This multiplicative update can be derived as a generalization of those for nonnegative and (singly) stochastic matrix factorization (Lee & Seung, 1999; Saul & Pereira, 1997). The main generalization is to introduce two sets of Lagrange multipliers into the update; these two sets are needed to enforce sum-to-one constraints on both the rows *and* columns of doubly stochastic matrices. The resulting update is similar but not equivalent to the Sinkhorn-Knopp procedure for projecting a nonnegative matrix onto the set of doubly stochastic matrices (Sinkhorn & Knopp, 1967).

The multiplicative updates in eq. (20) can be used to optimize the score in eq. (2), and unlike the Frank-Wolfe updates, they do not involve the expense of computing a projected gradient, as in eq. (9). But to use these updates on dense doubly stochastic matrices, it is necessary to compute the score in eq. (2) and the gradient in eq. (4) when $P$ is dense. Naively this appears to require $O(n^4)$ operations, a prohibitive scaling for matrices of size $n = 18524$. We devised a faster way to compute these gradients by exploiting the fact that the connectome weights are *quantized*. This method is described in Appendix B.

| Submitted | Name | Score |
|-----------|------|-------|
| 2025-02-18 | **Old School** | 5,853,925 |
| 2025-02-17 | D. A. Bader, H. A. Sriram, S. Chinthalapudi, and Z. Du | 5,853,910 |
| 2025-01-31 | **Old School** *(Winner)* | 5,853,779 |
| 2025-01-31 | D. A. Bader, H. A. Sriram, S. Chinthalapudi, and Z. Du | 5,849,534 |
| 2025-01-31 | Y. Ma, X. Zhu, and L. Zhu | 5,842,347 |
| 2025-01-31 | W. B. Hayes, M. Longo, and R. Longo | 5,841,041 |
| 2025-01-31 | Team FAQ | 5,838,188 |
| 2025-01-31 | D. Hashorva | 5,837,872 |
| 2025-01-31 | P. J. C. Duarte, C. Larsen, and R. Willemsen | 5,834,246 |
| 2025-01-31 | T. M. da Nóbrega | 5,824,339 |

Table 1: Top ten scores on the leaderboard of the VNC matching challenge as of December 2025. The winning score on 2025-01-31 was obtained using the methods in this paper, and the top score on 2025-02-18 was obtained by combining the methods of the two leading teams.

### 4.3 Comparison to competing methods

The official winning score to the challenge was 5853779. This score was submitted on January 31, 2025 and achieved by alternating the additive updates for sparse doubly stochastic matrices in eq. (11) with the multiplicative updates for dense doubly stochastic matrices in eq. (20). Two higher scores were subsequently obtained in February 2025, but only by using the winning solution as an initial seed. Table 1 reproduces the top ten scores on the leaderboard as of December 2025.

A number of competing methods for graph matching were explored by other teams in the challenge. None of these teams obtained scores above 5850K, indicated by the dotted line in Fig. 2, despite in some cases significantly more compute (e.g., hours to weeks versus minutes). The second-place team pursued a purely combinatorial search in a high-performance computing environment (Koblentz, 2025), and their ceiling suggests the limitation of this strategy even when backed by considerable resources. This approach, however, was able to *improve* on the winning solution (using it as an initial seed) after the conclusion of the challenge; this is consistent with our finding that combinatorial methods have a larger role to play than continuous ones in the later stages of search. The fourth-place team in the challenge obtained a score of 5841K with a mature package for biological network alignment based on simulated annealing (Mamano & Hayes, 2017). The strength of this package is its versatility, as it can be applied out of the box to optimize any objective function for graph matching. However, it is much slower in its initial stages than the Frank-Wolfe updates from section 3, which surpass its score within a few iterations. The fifth-place team obtained a score of 5838K with a method for graph matching based on fast approximate quadratic (FAQ) programming (Vogelstein et al., 2015). Their method, like ours, was based on a continuous relaxation of the problem, but it approximated the weighted-intersection score in eq. (1) by a simpler correlation score whose gradient is easier to compute. Their ceiling suggests the limitation of a strategy that optimizes an approximation to the weighted-intersection score rather than eq. (1) itself.

Overall the performances of these competing methods serve to reinforce the essential components of our approach. First and foremost to our success was the alternation of continuous and discrete-combinatorial methods for search. But the alternating search only unlocked the potential for higher-scoring matches; it required further effort to find these matches in an efficient manner. Other essential components of our approach include the optimized routines in Algorithm 1 for evaluating swaps and computing gradients, the heuristic in eqs. (16–17) for fast preconditioning in the linear assignment problem, and finally the optimizations described in section 4.2 for the last iotas of improvement.

### 4.4 Other related work

There is a large literature on continuous relaxations for graph-matching problems. We briefly mention a few other well-known approaches and discuss their suitability for the VNC matching challenge. Gold & Rangarajan (1996) investigated a graduated-assignment algorithm that combines softmax operations,

Sinkhorn rescalings, and an outer annealing procedure to optimize over doubly stochastic matrices and permutation matrices. Their framework can be applied to general quadratic assignment problems, but the softmax computations can be expensive for large dense matrices. More recently, Xu et al. (2019) studied how to align graphs by defining a *Gromov-Wasserstein* discrepancy and *minimizing* the non-convex loss

$$L(P) = \sum_{ijkl} (A_{ij} - B_{kl})^2 P_{ik} P_{jl}, \tag{21}$$

where $P$ is doubly stochastic. They minimize the loss in eq. (21) by a proximal gradient method; specifically they use the Sinkhorn-Knopp algorithm to solve a sequence of convex (linearized and regularized) subproblems, and they develop a recursive, divide-and-conquer procedure for larger graphs. They obtain state-of-the-art results on these problems, but they also report that the proximal-gradient method can be sensitive to the choice of regularization hyperparameter and require many iterations in large graphs.

Many authors have studied graph-matching problems for the special case of *symmetric* graphs with *least-squares* loss functions. There are strategies in this setting that exploit the simpler choice of loss function as well as the undirected structure of the graphs. Most notably, in this setting the continuous relaxation to doubly stochastic matrices yields a *convex* optimization with loss

$$L(P) = \|AP - PB\|_F^2. \tag{22}$$

Zaslavskiy et al. (2009) observed that the best-performing doubly stochastic matrices for this loss function do not project to the best-performing permutation matrices; this is the same difficulty that we observed in Section 3.2. To find better projections, they use solutions from the convex relaxation to initialize a path-following algorithm that solves a sequence of convex-concave programs. In later work, Lu et al. (2016) investigated a fast projected fixed-point algorithm for these types of graph-matching problems. They obtain significant speedups by avoiding the bottleneck of Hungarian algorithms on large graphs; this is the same bottleneck that we addressed via the heuristics in section 3.3. More recently, (Hermanns et al., 2023) pursued an altogether different strategy that lifts the problem of node alignment to the more general problem of aligning functions on graphs. Their algorithm starts by constructing graph Laplacians from the adjacency matrices $A$ and $B$ and finding an orthogonal matrix that aligns the bases defined by their leading eigenvectors. It then solves a least-squares problem to compute a mapping matrix between time-parameterized heat kernels and finally obtains a permutation from this mapping matrix by solving a linear assignment problem.

Our approach in this paper borrows elements from many of these approaches (e.g., continuous relaxations, convex subproblems, perfect matchings). As mentioned previously, though, the graphs in the VNC matching challenge are asymmetric as well as substantially larger (with millions of edges) than previous benchmarks. For the best results, the optimizations must also be tailored to the weighted intersection score in eq. (1).

## 5 Discussion

In this paper we have described the AC⊕DC optimizations behind the winning solution to the VNC Matching Challenge. The graphs in this challenge were large enough to foil exhaustive methods, but small enough to experiment with many different approaches on a modestly equipped computer. The highest-scoring solution was obtained by combining continuous relaxations, additive and multiplicative updates, bespoke graph decompositions, and higher-order swaps. But most of the work was done by alternating simple (but aggressively optimized) methods for continuous and discrete combinatorial search and exploiting the particular structure of the weighted-intersection score in eq. (1).

We mention several directions for future work. First, not all of the methods in this paper scale gracefully to larger graphs with hundreds of thousands of nodes. For such graphs, it seems necessary to develop divide-and-conquer methods that do not require the storage of dense $n \times n$ matrices where $n \gg 10^4$. Second, we expect the linear assignment problem in eq. (9) to remain a crucial subroutine for higher-order assignment problems (or at least for any problem whose score function can be linearized). We need to understand better why certain heuristics, such as the preconditioner in Section 3.3, lead to faster solutions, and then perhaps we can use this understanding to develop even faster approaches. Third, the winning solution to

the VNC matching challenge was implemented in MATLAB, a relatively high-level programming language, but it is surely possible to produce faster implementations that are better at exploiting sparsity, managing high-speed memory, and harnessing GPUs. Finally, the challenge was focused on connectomic graphs that are highly structured and biologically related. While our experiments with perturbed edge weights suggest that AC⊕DC is robust to moderate noise, it remains to study the algorithm systematically across different noise models and graph families.

Our work also provides lessons beyond the specific problem of graph matching considered here. One of the goals of the challenge was to determine if the graph structure of the VNC connectomes sufficed by itself to derive a high-scoring match. For this reason, the teams were not provided with auxiliary *node features* for each neuron (e.g., cell type, left/right hemisphere), let alone training examples of fully or partially matched graphs. But the methods in this paper could potentially be integrated into graph-matching frameworks where node features and/or labeled examples are available for richer forms of graph-based learning (Li et al., 2019; Haller et al., 2022; Zhou et al., 2023; Ramachandran et al., 2024). Even more generally, the alternating strategy explored here could be applied to a broader class of large-scale problems in combinatorial optimization. Many problems in discrete optimization and structured prediction admit continuous relaxations where gradient-based methods may converge to interior-point solutions. For such problems, our results suggest that it can be much more effective to alternate between discrete and continuous approaches rather than to pursue these individual approaches in isolation.

### Acknowledgements

The authors thank the anonymous reviewers of this manuscript for many useful pointers and suggestions. DL and LS are grateful to Sebastian Seung for introducing us to this problem. DL also acknowledges support by the Institute of Information & Communications Technology Planning & Evaluation (IITP) grant by the Korean MSIT (No. RS-2024-00457882, National AI Research Lab Project). AM, one of the organizers of the FlyWire VNC Matching Challenge, joined this work at the invitation of DL and LS after the conclusion of the challenge. DL and LS developed the methods described here, while the challenge data, evaluation procedures, and results were fixed prior to this collaboration. The authors are also grateful to Mala Murthy and Amy Sterling for their part in organizing the concluding workshop in March 2025 at Princeton University and to the other teams in the challenge who shared their results at this workshop.

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

## A  Preconditioner for perfect matching

In this appendix we provide further details on the method used to compute perfect matchings in section 3.3. To recall, a perfect matching is computed prior to each line search in order to find the permutation matrix $Q_t$ with the largest projection onto the gradient $\nabla S(P_t)$; see eq. (9). In this calculation, it is more convenient to consider how $Q_t$ is changed relative to the current permutation matrix $\Pi_t$ in eq. (13). To do so, we re-index the rows and columns of the gradient and equivalently seek the permutation $\omega$ whose corresponding matrix

has the largest projection onto the matrix $\Lambda = \nabla S(P_t)\, \Pi_t^\top$, as in eq. (16). Finally we use an internal matlab routine to compute the perfect matching $\omega$, as in eq. (18). This routine can be called most simply as

$$\omega = \texttt{matlab.internal.graph.perfectMatching}(-\Lambda), \tag{23}$$

but for the graphs arising from the VNC Matching Challenge, it takes roughly 15 minutes to compute each perfect matching in this way. In what follows, we describe two heuristics that we investigated to accelerate these matchings.

The heuristics stem from the observation that perfect matchings are not affected by adding constant rows or columns to the cost matrix that appears as an argument in eq. (23). In practice, we found that the matching routine executed much more quickly if the rows and columns of the nonnegative matrix $\Lambda$ were shifted to create strongly negative entries off the diagonal. We investigated two such shifts—one that has the effect of *zeroing* the diagonal elements, and another that has the effect (in general) of *decreasing* the diagonal elements but not necessarily zeroing them. For the *diagonal-zeroing* preconditioner, we perform the shift

$$\Lambda_{ij} \leftarrow \Lambda_{ij} - \tfrac{1}{2}(\Lambda_{ii} + \Lambda_{jj}), \tag{24}$$

while for the *diagonal-decreasing* preconditioner, we perform the shift

$$\Lambda_{ij} \leftarrow \Lambda_{ij} - \sum_{k \neq i}(\Lambda_{ik} + \Lambda_{kj}). \tag{25}$$

Note that eq. (25) is an equivalent way of writing eq. (17) in section 3.3, and it has no effect if the matrix $\Lambda$ is purely diagonal. Since $\nabla S(P_t)$ is a nonnegative matrix, so is the matrix $\Lambda$ in eq. (16), and as a result, both these shifts work to create strongly negative entries off the diagonal. Such a pattern suggests that the perfect match is close to an identity permutation, and in practice this was found to accelerate the internal MATLAB routine by a significant amount.

Fig. 3 shows the speedups provided by these preconditioners when they were used for the VNC matching challenge—roughly $5\times$ for the diagonal-zeroing shift in eq. (24), and roughly $50\times$ for the diagonal-decreasing shift in eq. (25). In particular, the latter reduced the time required for each perfect matching from 10-15 minutes to 10-20 seconds. In theory, these preconditioners should not affect the results returned by the perfect matching routine, and for small problems, we observed this to be true. However, for the large graphs in the VNC matching challenge, the routine did often return matches that were very slightly different for different preconditioners.

## B Cost and gradient computation for multiplicative updates

In this appendix we describe how to compute the alignment score in eq. (2) and its gradient in eq. (4) more efficiently for *dense* doubly stochastic matrices $P$. Our key trick is to exploit the fact that the connectome weights are *quantized*. In particular, each nonzero weight records a positive number of synapses, and therefore not only are the elements of $A$ and $B$ quantized, but so are the possible values of $\min(A_{ij}, B_{kl})$ in eq. (4). Let $\mathcal{Q} = \{q_0, q_1, \ldots, q_M\}$ denote the set of these quantized values, with $q_0 = 0$ and $q_i < q_{i+1}$, and let $\Theta(\cdot)$ denote the step function defined by $\Theta(z) = 1$ if $z > 0$ and $\Theta(z) = 0$ otherwise. Then it follows that

$$\min(A_{ij}, B_{kl}) = \sum_{m=0}^{M-1} \Theta(A_{ij} - q_m)\, \Theta(B_{kl} - q_m)\, (q_{m+1} - q_m) \tag{26}$$

for all connectome weights $A_{ij}$ and $B_{kl}$. Note how this identity expresses the minimum as a sum over $M$ components. We now use this identity to more efficiently compute the score in eq. (2) and the gradient in eq. (4). To do so, for each interval $(q_m, q_{m+1})$, we define connectome *components* with weights

$$A_{ij}^{(m)} = \Theta(A_{ij} - q_m)\, \sqrt{q_{m+1} - q_m}, \tag{27}$$

$$B_{ij}^{(m)} = \Theta(B_{ij} - q_m)\, \sqrt{q_{m+1} - q_m}. \tag{28}$$

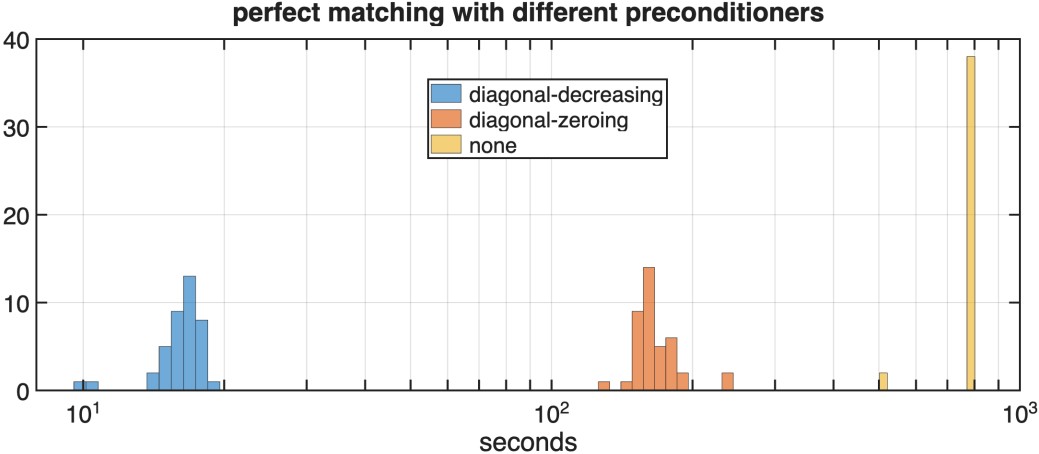

Figure 3: Effect of different preconditioners on the time required for MATLAB's internal graph routine for perfect matching. The figure shows histograms for each preconditioner; each histogram profiles the execution times of the perfect matching routine throughout the optimization in Fig. 1 (right). The diagonal-zeroing preconditioner in eq. (24) leads to a roughly 5× speedup, while the diagonal-decreasing preconditioner in eqs. (17) and (25) leads to a roughly 50× speedup.

Note that each connectome component is a sparse matrix in its own right, one that is at least as sparse as the connectome from which it is derived. Finally, combining eqs. (26–28), we rewrite the score in eq. (2) as

$$\mathcal{S}(P) = \sum_{ijkl} \min(A_{ij}, B_{kl}) P_{ik} P_{jl} \tag{29}$$

$$= \sum_{ijkl} \left[ \sum_m A_{ij}^{(m)} B_{kl}^{(m)} \right] P_{ik} P_{jl} \tag{30}$$

$$= \sum_m \text{trace} \left[ \left( A^{(m)} P \right) \left( P B^{(m)} \right)^\top \right]. \tag{31}$$

Note that this final expression for the score in eq. (31) can be computed in $O(Mn^3)$ as opposed to $O(n^4)$. This savings is significant when $M \ll n$, and it is also inherited by the computation of the gradient. For the VNC matching challenge, there are $M = 617$ graph components that arise from the nonzero connectome weights of the male and female fruit fly. With this savings, and using a GPU, it takes less than 2 minutes to compute the gradient of eq. (31) and perform each multiplicative update in eq. (20).

