# OpenReview forum: "AC$\oplus$DC search: behind the winning solution to the FlyWire graph-matching challenge"
_TMLR — Accepted by TMLR_

### Review · Reviewer_d4Zf · 2025-09-08

**Summary Of Contributions:**

# Summary

This paper presents the winning solution to the Flywire Ventral Nerve Cord Matching Challenge, which involved aligning connectomes of male and female fruit flies.  The authors detail two main optimization components for graph matching between large fly connectomes: (1) a greedy combinatorial permutation search, and (2) a continuous relaxation using Frank-Wolfe optimization over doubly stochastic matrices, enhanced by a preconditioning heuristic.

# Pros
* Addresses a real, large-scale problem with practical constraints
* Clear algorithmic exposition with detailed complexity analysis
* Novel combination of existing techniques yielding superior performance

# Cons
* Limited theoretical analysis of why the alternating approach works
* Some ad-hoc heuristics (especially the preconditioning) lack formal justification.
* Scalability concerns for much larger graphs are not thoroughly addressed.
* No direct, systematic comparisons with state-of-the-art graph matching algorithms

**Additional Comments:**

N/A

**Audience:**

Yes

**Audience Explanation:**

The emphasis on achieving strong results with modest computational resources aligns with practical ML concerns about accessibility and efficiency.

**Broader Impact Concerns:**

The work focuses on computational methods for biological network analysis, which is generally beneficial for scientific understanding. The specific application to fruit fly connectomes is purely for scientific research.

**Claims And Evidence:**

No

**Claims Explanation:**

Please provide more experiments to support the proposed model.

**Requested Changes:**

* Could the authors provide quantitative benchmarks or ablations showing the acceleration benefit from their preconditioning heuristic relative to standard assignment solvers and/or state-of-the-art alternatives?
* How would the approach behave under different forms of graph noise, weight quantization levels, or in settings where the input graphs are less similar?

---

### Review · Reviewer_1eku · 2025-10-02

**Summary Of Contributions:**

This paper proposes an approach for solving a graph matching problem. It focuses in particular on a problem that was given as a research competition, consisting in aligning the connectomes of a neural structure (ventral nerve cord) between two flies (male and female). The challenge had 18k nodes in both graphs.

The authors propose the approach that won the challenge. This approach relies on alternating greedy permutations in the matching matrix (as a discrete optim problem) and optimizing a continuous relaxation of the matching matrix (in the space of doubly stochastic matrices).

The paper's claims is that their approach provides good performances in a short time

**Additional Comments:**

I believe that this is an overall well written paper, with clear and detailed presentation of the method. I believe that the method in question, ACDC, is of interest to the community and proved its effectiveness in the FlyWire challenge. The components of the method are not novel (greedy swaps + Frank-Wolfe), but the paper succeeds in demonstrating that using them in an alternating fashion works better than each of them individually.

My main concern is the lack of comparison to other methods in terms of complexity/runtime, both methods used by other competitors but also more general methods for graph matching, since it is a well studied problem, even on large graphs. Moreover, it would be of great interest to the community to study the power of the method outside of the single, instanced problem posed by the competition. Analysis of the performance on a wider scope of settings could show the strengths and weaknessess of the method much further.

**Audience:**

Yes

**Audience Explanation:**

Yes, I believe that the winning solution to a graph matching competition anchored in biologic data is of interest to the respective communities of the methods (graph learning/graph matching) and applications (neuronal biologists).

**Broader Impact Concerns:**

No concern.

**Claims And Evidence:**

Yes

**Claims Explanation:**

Yes, I do believe that the paper provides clear and accurate evidence of their claims. The code is provided, and I mention here that I did not run the code to verify it.

The descriptions of the approach are clearly detailed, allowing to understand the processes and the grounds for decision. The ablation studies (fig 1) isolating both components also offers relevant insight into the method's motivation.

**Requested Changes:**

The scope and novelty of the paper are somewhat limited. It applies existing techniques to a single use-case, which provides no guarantee of generalization of the method with regards to other scenarios in the broad world of graph matching. I believe that comparing the approach to existing techniques on existing benchmarks would be very beneficial, as the community could benefit from a more exhaustive study of the strengths and weaknesses of the method. Please note that I fully acknowledge that this is not ground for rejection of this paper.

Moreover, the comparison to baselines is shallow, and there is a lack of comparison to other graph matching methods. The leaderboard of the competition is relevant in proving the method is performant, but a few issues remain:
* D. A. Bader, H. A. Sriram, S. Chinthalapudi, and Z. Du obtain very close results (albeit after the end of the competition). It would be relevant to discuss what method that team used and compare it to the ACDC approoach
* The runtime of the approach is evoked several time throughout the paper, but no comparison is offered with other methods from the leaderboard. How quick/slow is the Bader approach, for instance?
* The lack of related works is hindering a good situation of the approach in the larger scope. It is in my opinion necessary that other methods for graph matching be discussed and compared, especially those that were used during that competition and that achieved a good score. This would allow to understand the strengths of ACDC much better, and understand the generalization possibilities.

---

### Review · Reviewer_VkRf · 2025-10-12

**Summary Of Contributions:**

The paper describes the winning strategy of the Flywire Ventral Nerve Cord Matching Challenge. The proposed approach, called AC+DC, is based on alternating combinatorial and continuous optimization procedures. In essence, the problem consists of graph matching. AC+DC employs a local search based on pairwise swaps (i.e., pairs of indices in a permutation) and a relaxed formulation of the problem, in which doubly stochastic matrices naturally arise. The combination of these procedures led to the winning solution. The paper also provides a detailed discussion of the computational and algorithmic aspects underlying the method’s success.

**Audience:**

Yes

**Audience Explanation:**

I believe that members of the Graph Machine Learning community may find the proposed solution for graph matching particularly relevant. In particular, graph matching has become increasingly important for tasks such as graph similarity computation, cross-graph alignment, and subgraph retrieval, which are central to applications in molecule analysis, vision, and relational reasoning (e.g., see [1,2,3]).

[1] Improving Graph Matching with Positional Reconstruction Encoder-Decoder Network, NeurIPS 2023.

[2] Graph Matching Networks for Learning the Similarity of Graph Structured Objects, ICML 2019.

[3] Iteratively Refined Early Interaction Alignment for Subgraph Matching based Graph Retrieval, NeurIPS 2024.

**Broader Impact Concerns:**

I have no broader impact concerns.

**Claims And Evidence:**

Yes

**Claims Explanation:**

The paper is entirely focused on the specific problem of the VNC Matching Challenge, demonstrating that the proposed solution achieves top performance in this setting. Moreover, the paper is clearly written and provides comprehensive implementation details, including pseudocode.

**Requested Changes:**

I believe the paper would benefit from:

- Discussing the broader relevance to the ML community. It would be valuable to elaborate on how the ideas behind AC+DC could inform or inspire future developments in machine learning;

- Clarifying the motivation for the heuristics used to accelerate the perfectMatching routine. The rationale behind this component was somewhat unclear. Was it primarily introduced to avoid trivial identity permutations? How did the authors come to Eqs (16)-(19)?

- Including ablation studies to demonstrate the impact of the main design choices—possibly on smaller subproblems. Overall, I missed additional experiments that could justify the specific architectural and algorithmic decisions behind AC+DC, especially when compared to simpler or “vanilla” baselines;

- Expanding the discussion of related work. It remains unclear which prior methods have explored similar alternating combinatorial–continuous optimization strategies. A more detailed discussion would help contextualize the contribution and clarify the novelty of the proposed approach.

---

### Decision · Action_Editor_Q2SW · 2025-11-26

**Recommendation:** Accept with minor revision

**Additional Comments:**

This paper was reviewed by three reviewers. The reviewers raised concerns and requested corresponding changes regarding several aspects of the submission. These included the discussion of related work, the method's relevance to the machine learning community, the lack of comparisons against baselines in terms of performance and running time, the absence of ablation studies to evaluate key design choices (e.g., the preconditioning heuristic), and the method's performance under noise or different setups. The authors responded to the reviewers and promised to incorporate most of the requested changes in the revised manuscript. Following the authors' response, two reviewers recommended acceptance of the paper, while one reviewer recommended weak acceptance. Given that the paper has not yet been revised, I recommend a "minor revision" and I request that the final version of the manuscript implements the changes requested by reviewers and carefully considers the reviewers' comments.

**Audience:**

Yes

**Audience Explanation:**

Graph matching lies at the heart of several graph learning problems. Therefore, I believe that researchers in the graph machine learning community would be interested in the findings of this paper.

**Claims And Evidence:**

Yes

**Claims Explanation:**

The paper claims to present the winning solution to the Flywire Ventral Nerve Cord Matching Challenge. The proposed approach, which combines a combinatorial optimization over the symmetric group of permutations with a continuous relaxation of this problem to the space of doubly stochastic matrices, is clearly described in the manuscript. The paper also claims that the MATLAB implementation of the proposed approach can achieve a winning score to the challenge in less than 15 minutes on a laptop computer. This claim is supported by Figure 2.